# Postoperative Management in Patients with Pheochromocytoma and Paraganglioma

**DOI:** 10.3390/cancers11070936

**Published:** 2019-07-03

**Authors:** Divya Mamilla, Katherine A. Araque, Alessandra Brofferio, Melissa K. Gonzales, James N. Sullivan, Naris Nilubol, Karel Pacak

**Affiliations:** 1Section on Medical Neuroendocrinology, *Eunice Kennedy Shriver*, National Institute of Child Health and Human Development, National Institutes of Health, Bethesda, MD 20892, USA; 2Adult Endocrinology Department, National Institute of Diabetes and Digestive and Kidney Diseases, National Institutes of Health, Bethesda, MD 20892, USA; 3Cardiovascular and Pulmonary Branch, National Heart, Lung, and Blood Institute, National Institutes of Health, Bethesda, MD 20892, USA; 4Department of Anesthesiology, University of Nebraska Medical Center, Omaha, NE 68198, USA; 5Endocrine Oncology Branch, National Cancer Institute, National Institutes of Health, Bethesda, MD 20892, USA

**Keywords:** postoperative, pheochromocytoma, hypertension, hypotension, arrhythmia

## Abstract

Pheochromocytomas and paragangliomas (PPGLs) are rare catecholamine-secreting neuroendocrine tumors of the adrenal medulla and sympathetic/parasympathetic ganglion cells, respectively. Excessive release of catecholamines leads to episodic symptoms and signs of PPGL, which include hypertension, headache, palpitations, and diaphoresis. Intraoperatively, large amounts of catecholamines are released into the bloodstream through handling and manipulation of the tumor(s). In contrast, there could also be an abrupt decline in catecholamine levels after tumor resection. Because of such binary manifestations of PPGL, patients may develop perplexing and substantially devastating cardiovascular complications during the perioperative period. These complications include hypertension, hypotension, arrhythmias, myocardial infarction, heart failure, and cerebrovascular accident. Other complications seen in the postoperative period include fever, hypoglycemia, cortisol deficiency, urinary retention, etc. In the interest of safe patient care, such emergencies require precise diagnosis and treatment. Surgeons, anesthesiologists, and intensivists must be aware of the clinical manifestations and complications associated with a sudden increase or decrease in catecholamine levels and should work closely together to be able to provide appropriate management to minimize morbidity and mortality associated with PPGLs.

## 1. Introduction

Pheochromocytomas (PHEOs) are rare catecholamine-secreting neuroendocrine tumors derived from chromaffin cells of the adrenal medulla (80–85%). Paragangliomas (PGLs) are extra-adrenal tumors, originating from similar cells present in both sympathetic and parasympathetic ganglion cells (15–20%) [1]. Catecholamines, i.e., epinephrine (EPI), norepinephrine (NE) and dopamine (DA), are synthesized and secreted from almost all pheochromocytomas and paragangliomas (PPGLs) [2,3]. They are either released in large amounts during tumor manipulation or may suddenly drop after tumor resection causing wide swings in hemodynamics [4]. Clinical manifestations of fluctuating perioperative and postoperative hemodynamics are hypertensive crisis, arrhythmias (most commonly tachyarrhythmias), headache, sweating, constipation and anxiety. Therefore, attention should be focused on minimizing tumor manipulation by careful handling of tumor tissue, limiting intra-abdominal pressure, providing adequate anesthesia and maximizing the use of vasoactive agents to achieve intraoperative hemodynamic stability, hence improving outcomes during the postoperative period [4,5]. The use of an appropriate preoperative antihypertensive regimen can be counterproductive when effects continue after surgical tumor removal, i.e., a rapid decline of catecholamine levels may lead to hypotension [6,7]. As such, administration of volume expanders and vasopressor management would be critical in reversing vascular collapse [8]. Reports have shown that patients with higher preoperative metanephrines and catecholamines have higher postoperative complications including organ ischemia, bowel obstruction, hypoglycemia, etc. [9,10,11].

Since the pioneering work by Gagner et al. of the first laparoscopic resection of PHEO in 1992, surgical management of PHEO has considerably improved owing to the advancement in pre, intra, and postoperative care of these patients [12]. Postoperatively, patients are closely monitored in the intensive care unit for hemodynamic fluctuations along with a careful assessment of electrolyte and endocrine abnormalities.

To our best knowledge, most of the published articles focus primarily on preoperative and intraoperative care of PPGL patients, whereas studies detailing postoperative management are only available from individual case reports. It is extremely important for physicians of PPGL patients to provide not only appropriate preoperative evaluation and treatment but also adequate postoperative care.

In this article, we describe the medical approaches to treat these patients after tumor resection based on our unique, long-standing experience with these patients at the National Institutes of Health. Additionally, we present the notable complications physicians should become aware of, including those emergencies that require immediate attention by a well-trained and experienced endocrinologist working alongside intensivists. Finally, this article provides clinical caveats to practicing clinicians regarding postoperative management of these patients.

## 2. Catecholamines and Adrenoceptors

PPGLs secrete catecholamines with substantial variation in their content based on the expression of various biosynthetic enzymes. Typically, adrenal PHEOs produce either EPI or NE while extra-adrenal and metastatic PHEOs mainly produce NE. Rarely, these tumor cells produce DA. Adrenoceptors (α_1_, α_2_, β_1_, β_2_) are the final target site of action for these catecholamines. Therefore, it is essential to recognize the impact of catecholamines from PPGL on specific organs (Table 1 and Table 2).

EPI and NE have some overlapping but distinct effects on α- and β-adrenoceptors. EPI has more potent effects on β_2_-adrenoceptors than NE, but equivalent effects on β_1_-adrenoceptor, along with dominant effects on α-adrenoceptors in comparison to NE (Table 1). More than 95% of EPI is released from the adrenal medulla, which acts on the β_2_-adrenoceptors of skeletal muscle vasculature causing vasodilation leading to hypotension. In contrast, NE released from sympathetic nerve endings within the effector sites causes α_1_-adrenoceptor mediated vasoconstriction, leading to hypertension and its profound action on β_1_-adrenoceptors causes increased ionotropic and chronotropic effects in the heart [14]. Eventually, the resulting concentration of catecholamines at effector sites are significant determinants of adrenoceptor mediated responses [14,15].

Persistently high catecholamine levels may lead to adrenoceptor desensitization due to receptors’ internalization, reduction of their numbers on the cell surface, or decreased binding affinity of catecholamines to receptors [16,17]. These mechanisms may partially explain why some patients with PPGL are only moderately hypertensive, despite high plasma catecholamine levels. Decreased or desensitized adrenoceptors, perioperative α-adrenoceptor blockade, and abruptly decreased catecholamine production contributes to postoperative hypotension.

Differences in receptor binding, affinity, and downstream effects explain the spectrum of clinical signs that patients with PPGL develop. Patients with predominantly EPI secreting PPGL have episodic symptoms and signs of palpitation, lightheadedness or syncope, anxiety, and hyperglycemia. Conversely, patients with primarily NE secreting tumors have continuous symptoms and signs of hypertension, sweating, and headache [18,19,20]. These effects could extend to peri and postoperative periods due to excessive catecholamines release while handling the tumor during surgical resection. Thus, it is mandatory that these patients are followed by a multidisciplinary team in a close monitoring setting like the intensive care unit during the postoperative period.

## 3. Cardiovascular Complications Related to PPGLs

### 3.1. Hypertension

According to the American College of Cardiology/American Heart Association (ACC/AHA) 2017, hypertension is defined as blood pressure > 130/80 mmHg [21]. Risk factors associated with postoperative hypertension following PPGL resection include incompletely removed/metastatic tumor, additional tumor at an unknown location, underlying essential hypertension, excessive intravenous fluid administration, pain, and excessive use of vasopressors (management of transient hypotension resulting from a significant drop in catecholamine levels).

One possible mechanism behind hypertensive crisis during PPGL resection includes massive catecholamine release secondary to tumor manipulation. Watchful and precise resection, appropriate vasodilator use, and clear communication between the surgeons and anesthesiologists are helpful in minimizing intraoperative hemodynamic fluctuation [22,23,24]. Other tumor-related factors resulting in hemodynamic instability include large tumor size (>4 cm), urinary catecholamines > 2000 µg/24 h, large postural drop in blood pressure (>10 mmHg) after α-adrenoceptor blockade, preoperative mean arterial blood pressure (MAP) > 100 mmHg and prolonged duration of anesthesia [25,26,27]. Therefore, hypertension should be treated promptly with quickly titratable and shorter-acting agents. Below we summarize the most important therapeutic options available to treat perioperative hypertension among patients with PPGL (Figure 1) (Table 3).

#### 3.1.1. α- and β-Adrenoceptor Antagonist

α-adrenoceptor antagonists are categorized according to specific receptor activity: Non-selective antagonists such as phenoxybenzamine and phentolamine and selective α_1_-adrenoceptor antagonists such as doxazosin, prazosin, and terazosin. These are predominantly used to manage perioperative hypertension following PPGL resection [2,28]. However, hypertensive crisis in the intra and postoperative period is managed by using phentolamine (Regitine) given as a 5 mg bolus intravenously with additional bolus doses given as needed [29]. The most common side effect is reflex tachycardia caused by baroreceptor reflex after α_2_-adrenoceptor blockade. Therefore, it is recommended to be used in combination with esmolol (Brevibloc), a rapidly acting cardio-selective β_1_-adrenoceptor antagonist [30]. The negative inotropic and chronotropic effect with no direct vasodilatory action of esmolol makes a good pair with direct acting α-adrenoceptor antagonists [30]. It is given intravenously at a loading dose of 0.5–1.0 mg/kg over 30 s or one minute followed by 50–150 µg/kg/min infusion (lower dose and slower infusion for more gradual control, higher dose and faster infusion for immediate control) and, if necessary, the dose may be increased up to a maximum of 300 µg/kg/min [31,32]. At higher doses, esmolol inhibits β_2_-adrenoceptors located in the musculature of bronchi and blood vessels. Moreover, as patients’ oral intake improves, cardio-selective β_1_-adrenoceptor blockers such as metoprolol (Lopressor) 25–50 mg three to four times a day or atenolol (Tenormin) 12.5–25 mg once a day (occasionally twice daily) orally are also administered to control catecholamines or α-adrenoceptor blocker induced tachyarrhythmia [33,34].

Labetalol (Normodyne or Trandate) is a combined selective α_1_- and non-selective β-adrenoceptor blocker with α to β blocking ratio of 1:4–7 [35]. Prolonged duration of action (2–4 h) of labetalol makes it challenging to titrate as a continuous infusion [36]. Poopalalingam et al. reported the use of continuous infusion of labetalol during PHEO resection, thus, providing sufficient α- and β-adrenoceptor blockade for the duration of surgery including the immediate postoperative period. Additionally, labetalol also minimizes the risk of postoperative hypotension and somnolence associated with preoperative use of phenoxybenzamine for the surgical preparation of a patient [37]. Labetalol is given intravenously at a loading dose of 10–20 mg followed by doubling of the initial dose every 10 mins until target blood pressure is achieved. If continuous infusion is planned, the dose is started at 2 mg/min and the drip rate is adjusted depending on blood pressure response with a total dose of up to 300 mg (FDA dosage). Labetalol and the other β-adrenoceptor blockers are used with caution in patients with bradycardia, atrio-ventricular block, and it may potentiate the action of other drugs such as calcium channel blockers [38,39].

#### 3.1.2. Calcium Channel Blockers

Although calcium channel blockers (CCBs) are less effective than α- and β-adrenoceptor blockers, it is an important class of drug used to manage hypertension following PPGL resection. They minimize complications related to catecholamines overload, such as coronary vasospasm [40]. CCBs inhibit NE mediated transmembrane calcium influx into vascular smooth muscle, resulting in arterial vasodilation [41]. In a clinical setting, nicardipine (Cardene) is most commonly used either alone or in combination with α- and β-adrenoceptor blockers in the postoperative period. However, we might also consider using those CCBs such as amlodipine or nifedipine, when already used in the preoperative period to control blood pressure. Nicardipine is started at a dose of 5 mg/h and dose is increased by 2.5 mg/h every five minutes to a maximum of 15 mg/h until target blood pressure is achieved. Recently, the use of Clevidipine (Cleviprex), an ultrashort-acting third-generation dihydropyridine CCB, has become popular. It acts by selective arteriolar dilatation which helps in decreasing peripheral vascular resistance [42]. Clevidipine starting dose is 1–2 mg/h intravenously and doubled every 90 s with a maximal dose up to 32 mg/h for a maximum duration of 72 h. Once the target blood pressure is reached, infusion is titrated based on therapeutic response and usually, maintained at a rate of 4–6 mg/h [43]. Multiple clinical trials have demonstrated promising effects of Clevidipine to control both pre and postoperative hypertension [44,45].

#### 3.1.3. Nitroglycerin

Nitroglycerin is an antianginal drug and a more potent venous dilator than arterial dilator. However, arterial dilation occurs at higher doses [46]. It has a rapid onset of action, and the dose is easily titratable, thus, often used to manage hypertensive emergency [4]. Nitroglycerin is most commonly used to manage intraoperative hypertensive crisis due to its effective safety profile [8]. The initial dose of intravenous nitroglycerin is 5 µg/min, which can be increased to a maximum of 100 µg/min. Nitroglycerin should be administered cautiously among patients undergoing PPGL resection as it is associated with hypotension, reflex tachycardia, and headache.

#### 3.1.4. Hydralazine

Hydralazine (Apresoline) use is very limited in the management of postoperative hypertension following PPGL resection due to its long duration of action resulting in hypotension and reflex tachycardia [47]. Hydralazine is preferably used among those patients who have underlying essential hypertension. The initial dose of hydralazine for an acute hypertensive episode is 10 mg intravenously (not exceeding 40 mg) given over two minutes with additional doses given every four to six hours as needed [30]. If patient’s oral intake is improved, hydralazine can be given at a dose of 100–200 mg orally two to three times a day. Adverse effects include tachycardia, negative effect on myocardial metabolism precipitating acute myocardial ischemia or infarction, and cerebral vasodilation leading to increased intracranial pressure [47]. Thus, hydralazine is not considered a first-line agent to treat acute postoperative hypertension.

#### 3.1.5. Magnesium Sulfate

Magnesium (Sulfamag) is considered safe and potent supplemental medication to block catecholamine action, bringing hemodynamic control in adults and children undergoing PPGL resection [24]. Multiple case reports have verified the use of magnesium in pediatric age with PHEO [48,49,50]. The vasodilator property of magnesium is evoked by direct inhibition of catecholamine receptors and release of catecholamines from the adrenal medulla and adrenergic nerve terminals which act as endogenous calcium antagonists [51,52]. Given its safety profile and efficient blockade, it is considered a first-line agent in the intraoperative management of PPGL resection during pregnancy [53,54]. It is usually given at a dose of four grams in 250 mL of 5% Dextrose injection at a rate not exceeding 3 mL/min. Magnesium sulfate is also beneficial in the management of cardiac arrhythmias due to its stabilizing effect on cardiac electrical conduction [55]. Therefore, continuous infusion of magnesium is given under close monitoring due to its associated neuromuscular and cardiovascular toxicities especially among patients with renal insufficiency. These toxic complications can be reversed with the intravenous calcium administration.

#### 3.1.6. Treatment of Underlying Hypertension

Persistent hypertension following PPGL resection might be related to coexistence of essential hypertension. Oral antihypertensive medications, preferably a preoperative antihypertensive medication regimen, should be used under close monitoring to make any necessary adjustments in dosage. After discharge, a patient is followed by his or her primary care provider. Approximately six weeks later, laboratory evaluation is performed to evaluate for the presence of catecholamines and metanephrines in blood and urine, thus, identifying the presence of residual tumor, tumor at an unknown location, or metastatic tumor. However, if laboratory results are negative, a diagnosis of underlying essential hypertension is established, and antihypertensive medications are given as necessary.

### 3.2. Hypotension

Hypotension is defined as blood pressure below 90/60 mmHg or any degree of low blood pressure leading to organ hypoperfusion or end-organ damage. Potential risk factors attributed to postoperative hypotension following PPGL resection include chronically low circulating plasma volume, an abrupt decrease in serum catecholamine levels, down-regulation of adrenoceptors, increased blood loss, and cardiogenic or septic shock [56]. Independent tumor-related risk factors are open procedures, high preoperative plasma NE and EPI levels and increased urinary fractionated catecholamines excretion [25,26,57,58]. EPI secreting PPGLs cause decreased cardiac contractility owing to downregulation of β-adrenoceptors on the heart, resulting in left heart failure, thus precipitating hypotensive state and collapse after resection [59,60,61]. However, NE secreting PPGLs cause a decrease in circulating plasma volume secondary to α_1_-adrenoceptor mediated vasoconstriction [2]. Moreover, profound irreversible α-adrenoceptor blockade by phenoxybenzamine triggers recalcitrant postoperative hypotension by: (1) Prolonged half-life of the drug and (2) permanent covalent binding to adrenoceptors which are curtailed only after de novo synthesis [2,62]. Nevertheless, this effect is comparatively less with selective α_1_-adrenoceptor blockers.

First line management for postoperative hypotension following PPGL resection includes vigorous intravenous fluid administration [40]. If fluid therapy fails, vasopressor use is justified to restore normal blood pressure. Vasopressors used to manage hypotension include NE, EPI (rarely), and vasopressin. However, pure α-adrenoceptor agonists such as phenylephrine are not used because of remnant effects of preoperative α-adrenoceptor blockade. Finally, the ultimate goal of managing hypotension is both restoration of adequate and prevention of inadequate tissue perfusion (Figure 2).

#### 3.2.1. Intravenous Fluid

Fluid loss during the postoperative period results from: (1) Surgical site bleeding or occult bleeding, (2) suction drainage (nasogastric tube, chest tube for pleural fluid, ascites from abdominal drains and urinary system), (3) third spacing of fluid, i.e., capillary leak and extravasation of protein-rich serum into the interstitial spaces of the soft tissues, organs, deep space cavities (chest, abdomen), or retroperitoneum, (4) insensible or evaporative losses, and (5) loss of systemic vascular resistance after sudden withdrawal of catecholamines following tumor resection [63]. First line management for hypotension is giving adequate amounts of intravenous fluid to restore blood volume, blood pressure, and adequate tissue perfusion, by filling the large acute increase in volume of distribution thus, reducing morbidities associated with hypotension [64]. Blood transfusion is reserved for those patients who present with excessive operative site bleeding while monitoring serum hemoglobin levels at regular intervals and in patients with known coronary disease. Knowledge of both systolic and diastolic left ventricular function is essential to anticipate the possibility of volume overload and associated complications. Every patient should be monitored carefully for signs of volume overload since this may lead to postoperative hypertension and pulmonary edema leading to respiratory compromise. The use of vasopressors is justified only when there is persistent hypotension despite administration of adequate amounts of intravenous fluids. Patients who are hemodynamically at risk, i.e., low ejection fraction or known coronary disease can be monitored by echocardiography, pulmonary artery catheters or other non-invasive devices.

#### 3.2.2. Norepinephrine 

NE (Levophed) is an α- and β-adrenoceptor agonist with stronger affinity towards α_1_-adrenoceptor and mild to moderate β_1_-adrenoceptor and minimal β_2_-adrenoceptor agonist effects [65]. It is considered a first-line agent if intravenous fluid therapy fails to correct hypotension. The loading dose of NE is 8–12 µg/min intravenously with flow rate adjusted to maintain low normal blood pressure with mean arterial pressure >65 mm Hg. Maintenance dose is adjusted to 2–4 µg/min with daily doses as high as 68 mg. NE increases mean arterial pressure, effective circulating volume, venous return, and preload with minimal increase in heart rate [66]. As the dose of NE is increased within safety limits, improved fluid responsiveness can be seen due to β_1_-adrenoceptor mediated augmentation of venous return [67]. NE infusion is used in patients with adequate volume resuscitation, otherwise this may cause profound ischemia leading to serious adverse effects. Therefore, judicious use of NE is carefully monitored in postoperative PPGL patients due to altered hemodynamics caused by excessive release of catecholamines from tumor cells.

#### 3.2.3. Epinephrine 

EPI (adrenalin) is an α- and β-adrenoceptor agonist with dose-dependent actions. Mixed actions of epinephrine cause an increase in both mean arterial blood pressure by vasoconstriction (α_1_-adrenoceptor effect) and cardiac output (β_1_-adrenoceptor effect) [68,69]. The starting dose of infusion is 1–4 µg/min and can be titrated up by 1–2 µg/min every 20 min until the desired effect is achieved. Total infusion dose should not exceed >10 µg/min [70]. It is considered a second-line agent in the treatment of refractory hypotension [71,72]. Adverse effects associated with the use of EPI include pulmonary hypertension, tachyarrhythmia, myocardial ischemia, lactic acidosis, and hyperglycemia. Henceforth, EPI is cautiously used in PPGL resected patients since it can prolong the effects of catecholamines released from the tumor cells.

#### 3.2.4. Vasopressin

Vasopressin (pitressin, pressin), a nonapeptide hormone, is released from the posterior pituitary gland in response to increased plasma osmolality, decreased intravascular volume and low blood pressure. Vasopressin restores water balance and blood pressure by acting on vasopressin 1 (V_1_), and V_2_ receptors in the following ways: (1) Increased systemic vascular resistance by V_1_ receptor stimulation on arterial smooth muscle cells and (2) water reabsorption at collecting ducts by stimulation of V_2_ receptors in kidney. Additionally, vasopressin may increase responsiveness of catecholamines allowing lower doses of adrenergic drugs to be used [73]. Few patients develop catecholamine-resistant vasoplegia through severe catecholamine deficiency induced by tumor removal. In some cases, aggressive catecholamine and fluid replacement therapy might not be helpful to restore vascular tone. Augoustides et al. proposed that a chronic increase in NE levels can inhibit vasopressin release through downregulation of the neurohypophyseal vasopressin synthesis [74]. Vasopressin has no action on adrenergic receptors and is thus useful to treat refractory hypotension after PPGL resection [24]. In clinical practice, vasopressin is most often used at 0.03 to 0.04 units/min. Higher doses have been used but with a higher risk of adverse events.

### 3.3. Arrhythmia

Broadly, arrhythmias are classified as tachyarrhythmias (>100 beats/min) and bradyarrhythmias (<60 beats/min). Tachyarrhythmia is commonly documented following PPGL resection from either increased catecholamine levels or inotropes used to correct postoperative hypotension [75]. Other important risk factors include rebound effect of β-adrenoceptor blocker discontinuation and residual effect of α-adrenoceptor blockade used in preoperative preparation, postoperative hypotension, anemia and pain that also may cause an increase in sympathetic activity. Sinus tachycardia is the most commonly observed tachyarrhythmia after PPGL resection as a compensatory response to an underlying condition. Therefore, treatment is focused on addressing underlying causes, such as relieving pain or anxiety and replacing volume deficit. If heart rate is persistently elevated for a prolonged period of time, rate-controlling medications may be indicated. Other tachyarrhythmias observed following PPGL resection may include atrial fibrillation, atrial flutter, and occasionally life-threatening ventricular fibrillation. Management of tachyarrhythmia is based on a patient’s hemodynamic stability, presence or absence of narrow or wide complex QRS, and regular or irregular rhythm.

Residual anesthetic effects and opioids and analgesics used in the postoperative period makes it difficult to identify the signs and symptoms of unstable tachyarrhythmia. Therefore, a multidisciplinary team of physicians, including intensivists, surgeons, anesthetists, cardiologists, and endocrinologists are required to manage these patients.

In stable patients with narrow, regular (<0.12 sec) QRS complex that is not sinus tachycardia (non-sinus tachycardia), vagal maneuvers such as Valsalva or carotid sinus massage can be executed. If rhythm is regular, adenosine 6 mg rapid intravenous push is injected followed by a bolus dose of 12 mg when no response is observed within one to two minutes. An additional dose of 12 mg is given in resistant cases. Usually, adenosine is helpful to effectively terminate and/or diagnose tachyarrhythmia. If tachycardia resumes or is not responsive to adenosine, treatment with longer-acting atrioventricular (AV) nodal blocking agents such as diltiazem or β-adrenoceptor blocker is required. Stable patients with irregular narrow complex tachycardia usually have either atrial fibrillation or atrial flutter requiring an expert consultation and use of rate-controlling agents such as diltiazem, β-adrenoceptor blockers or antiarrhythmics such as amiodarone. Diltiazem (Cardizem) is given intravenously at a starting dose of 0.25 mg/kg over two minutes followed by a maintenance dose with an infusion at 5–15 mg/h. Commonly used β-adrenoceptor blockers to control tachyarrhythmia include metoprolol (intravenous and oral forms) and esmolol (very short half-life, existing in drip form). Esmolol (Brevibloc) is loaded initially as 500 µg/kg over one minute with maintenance doses of 50–200 µg/kg/min and repeat bolus doses between each dose increase. However, a major drawback of using esmolol is that arrhythmias may recur after discontinuation. Therefore, longer acting β-adrenoceptor blocker, metoprolol tartrate (Lopressor), is used and given at a bolus dose of 2.5–5.0 mg intravenously over two minutes with repeated doses every 10 min up to a maximum three times before using longer intervals (every six to eight hours) [76]. These medications must be given under careful supervision as they can cause a significant drop in blood pressure, resulting in decreased perfusion to vital organs. Once a patient’s oral intake improves, these medications can be changed to oral formulations such as metoprolol tartrate, given orally at a dose of 25–100 mg twice daily accordingly. Rhythm controlling agent such as amiodarone, an iodinated benzofuran antiarrhythmic may be considered. It is easily titratable and is used to treat both supraventricular and ventricular tachyarrhythmia with limited negative inotropic effects. Typical doses in stable patients includes a bolus of 150 mg intravenously over 10 min followed by an infusion of 1 mg/min over six hours which is titrated to 0.5 mg/h after the first six hours with a maximum dose of up to 2.2 g in the first 24 hrs. One large metanalysis done on the effect of amiodarone on recent-onset atrial fibrillation showed a similar efficacy and safety profile to other antiarrhythmics [77].

For hemodynamically stable patients with wide complex tachycardia, 12-lead electrocardiogram (EKG) and expert consultation are obtained. Amiodarone is often the first medication used if there are no contraindications (hypersensitivity, history of severe bradycardia) are observed [78]. Recommended doses are described earlier in this section. Other antiarrhythmics frequently used after obtaining appropriate expert consultation include propafenone, flecainide, and procainamide.

For prolonged, refractory or hemodynamically compromising tachyarrhythmias, electrical cardioversion is considered in combination with antiarrhythmics. If a patient has pulseless electrical activity [2], immediately follow the ACC/AHA treatment guidelines for pulseless electrical activity (PEA) algorithm to manage appropriately (Figure 3).

It is not unusual for sinus tachycardia to remain uncontrolled despite using all drugs including esmolol, metoprolol, and diltiazem. Therefore, another therapeutic approach may be considered. Recently, ivabradine has been introduced to treat patients with heart failure and sinus tachycardia who failed or were intolerant to β-adrenoceptor blockers. Ivabradine selectively inhibits I_f_ current in the sinus node, resulting in decreased heart rate without compromising myocardial contractility and cardiovascular hemodynamic status [79]. Thus, Ivabradine is prescribed at a dose of 2.5–7.5 mg two times a day orally to PPGL patients with medication-resistant sinus tachycardia, providing excellent control of tachycardia (personal observations). It is recommended to start Ivabradine at low doses and titrate to achieve heart rate control [80,81].

In contrast, bradyarrhythmia’s after PPGL resection might result from sinus node or atrioventricular node dysfunction caused by progression of native conduction disease, overdose of rate controlling agents, i.e., β-adrenoceptor blocker, and metabolic or electrolyte disturbances. Bradycardia is defined in postoperative PPGL patients as heart rate below 60 beats/min. Therapeutic intervention is unnecessary for cases of transient bradycardia with no hemodynamic instability [82]. Detrimental effects of sustained bradycardia in the postoperative period include hypotension and inadequate cardiac output (decreased coronary circulation leading to myocardial infarction). Thus, atropine or β-adrenoceptor agonists may be indicated [75,82]. Atropine is administered at a dose of 0.5 mg intravenously, repeated every three to five minutes to a total dose of 3 mg intravenous until target heart rate of >60 beats/min is achieved. If bradycardia remains uncontrolled, consider using DA or EPI. DA actions are mediated either directly or by conversion and release of NE, leading to the stimulation of β-adrenoceptors on the heart while EPI increases heart rate by directly stimulating β-adrenoceptors on the heart [83]. DA has a dose range of 2–20 µg/kg/min whereas EPI is given at a dose of 2–20 µg/min until the desired heart rate >60 beats/min is sustained. Other agents which can be utilized to chemically pace are isoproterenol and dobutamine.

Unusually, bradyarrhythmias are unresponsive to atropine, DA, and EPI. Henceforth, transcutaneous or transvenous pacing becomes necessary.

### 3.4. Myocardial Infarction

It is not uncommon to recognize myocardial injury following PPGLs resection. The mechanism behind injury to myocytes by increased catecholamines includes hemodynamic compromise (increased afterload), tachycardia, increased oxygen consumption, and coronary arterial vasoconstriction [84]. Moreover, myocardium is rich in paraganglionic fibers with higher NE affinity which might produce structural and functional remodeling in the heart [85]. Direct actions of increased EPI and NE levels may result in myocarditis or cardiomyopathy causing leakage of cardiac enzymes [86]. Perhaps, hypotension, hypertension, anemia, hypoxemia, systolic, and/or diastolic dysfunction are common risk factors associated with myocardial infarction in the postoperative period. Clinical presentation of myocardial infarction in the postoperative period is commonly asymptomatic or non-specific, thus, relying on classic symptoms alone might lead to missed diagnoses [87]. Moreover, ischemic electrocardiographic signs may be subtle, and angina is masked by residual anesthetic effect and use of stronger analgesics concealing the diagnosis of myocardial injury [88]. Therefore, a higher index of suspicion of acute myocardial infarction is necessary in the absence of classic coronary risk factors [89]. According to ACC/AHA 2014 perioperative clinical practice guidelines, may consider comprehensive risk assessment with 12-lead EKG, 2D-echocardiography, and exercise or non-pharmacological stress testing to allocate them to low-risk or high-risk groups [90,91]. Measurement of troponin levels and obtaining a 12-lead EKG is recommended among patients with signs and symptoms of myocardial ischemia or infarction in the postoperative period [91]. To manage patients appropriately, myocardial infarction in the postoperative period is classified similarly as in the non-surgical setting: (1) Non-ST-elevation myocardial infarction (NSTEMI) (Figure 4) (Table 4) and (2) ST-elevation myocardial infarction (STEMI), management is dependent on expert consultation.

### 3.5. Heart Failure

Heart failure may also be seen as one of the rare complications seen following PPGL resection. Myocardium may have been dependent on the excess catecholamines prior to resection due to desensitized adrenoceptors, possible catecholamine-induced cardiomyopathy or non-ischemic cardiomyopathy. Reduced circulatory volume may also play a significant role in the development of heart failure following tumor resection [92]. Therefore, patients in the postoperative period following PPGL resection might develop either heart failure with preserved ejection fraction (HFpEF) or heart failure with reduced ejection fraction (HFrEF). HFpEF is usually seen in the setting of hypertensive damage with diastolic dysfunction resulting in increased susceptibility to volume overload and resulting pulmonary edema. In contrast, HFrEF may be observed in patients with significant myocardial ischemia secondary to postoperative instability. Heart failure may also develop in the setting of stress-induced cardiomyopathy called Takotsubo cardiomyopathy or broken-heart syndrome, that has been frequently described in patients with PPGL. Diuretics and inotropes may be utilized to treat heart failure. However, when medical therapy fails, mechanical devices such as intra-aortic balloon pump, impella, extracorporeal membrane oxygenation may need to be utilized. It is appropriate to promptly obtain cardiology consultation when there is a suspicion of heart failure to apply the most relevant diagnostic and therapeutic tools.

### 3.6. Cerebrovascular Accident

Clinical course of patients with PPGL can be complicated by ischemic or hemorrhagic stroke. The main mechanisms include: (1) Thrombotic or embolic occlusion of cerebral artery secondary to increased platelet aggregation [93], (2) cerebral hypoperfusion secondary to tissue hypoxia causing increased susceptibility in the watershed regions [94,95], and (3) rupture of intracranial artery causing intracerebral or subarachnoid hemorrhage secondary to catecholamine-induced hypertension [96].

Patients with PPGL have a higher rate of ischemic stroke in comparison to hemorrhagic events in the postoperative period [97,98]. The potential risk factors for catecholamine-induced ischemic stroke include uncontrolled hypertension (failure of cerebrovascular autoregulation), vascular spasm, dilated cardiomyopathy with risk of left ventricular thrombus (embolic) or atrial fibrillation (embolic).

Symptoms may range from headaches to motor and/or sensory deficits, to confusion and seizures. Recognition of these symptoms is challenging due to the concomitant use of anesthetic or sedative therapy in the postoperative period [99]. The modified National Institute of Health Stroke Scale (mNIHSS) has improved reliability benefits and, therefore, should be used for detailed neurological examination of patients with increased risk of postoperative stroke [100].

For patients that develop ischemic or hemorrhagic stroke following PPGL resection, it is advisable to provide a multidisciplinary patient-centered treatment plan including neurology, critical care, surgery, interventional neuroradiology, and anesthesiology.

## 4. Other Complications

### 4.1. Adrenocortical Insufficiency

Surgical approaches for PHEO resection can lead to primary adrenal insufficiency (PAI) in order of ascending frequency: Cortical sparing vs. unilateral vs. bilateral adrenalectomy. Glucocorticoids supplementation is not routinely prescribed for patients undergoing unilateral adrenalectomy because the contralateral adrenal gland can maintain eucortisolemia [101]. An individualized preoperative evaluation to identify risk factors for primary or secondary adrenal insufficiency (AI) is recommended, which includes but are not limited to identification of exposure to exogenous glucocorticoids or other medications that inhibit steroidogenesis, suppression of hypothalamic pituitary adrenal (HPA) axis secondary to endogenous cortisol co-secretion, or patient-related comorbidities. Currently, cortical sparing surgical approach for PHEOs with germline mutations and low risk for metastasis have increased. Adrenal sparing surgery is also preferred in patients with either solitary adrenal gland or bilateral adrenal involvement, as chronic glucocorticoid replacement is associated with decreased quality of life, increased cardiovascular risk, fatigue, infections and decreased resistance to stress [102,103].

When adrenal insufficiency is clinically suspected among patients undergoing bilateral adrenalectomy or in cases of subclinical or overt cortisol co-secretion, hydrocortisone 50 mg intravenous bolus is administered before anesthesia induction followed by 25–50 mg intravenously every eight hours thereafter, with tapering over the next 24–48 h. If no immediate postoperative complications develop, and patient can tolerate oral intake, transition to physiological oral maintenance dose with hydrocortisone 10–12 mg x body surface area (BSA) is recommended [104,105]. The proposed dosages can be individualized based on the patient’s history and length of surgery. If immediate complications develop, patients should remain on supraphysiologic doses of steroids as per clinical judgment.

Clinical surveillance for symptoms suggestive of AI during the postoperative period is paramount. In addition, morning serum cortisol measurements with adrenocorticotropic hormone (ACTH) levels and Cosyntropin stimulation test are routinely used to confirm the diagnosis of AI [106]. Cosyntropin stimulation test can be performed as early as postoperative day 1 (unless precluded by clinical instability) to three to six weeks later to identify those non-critically ill patients who will benefit with glucocorticoid replacement therapy [107]. Once the diagnosis of AI is established, continuation of physiological glucocorticoid replacement is indicated. In cases of cortisol co-secretion or underlying central AI, HPA axis should be evaluated every six months to annually to assess for recovery. In contrast, after bilateral adrenalectomy, lifelong glucocorticoid and mineralocorticoid supplementation without further HPA axis testing is recommended [101,108,109].

Mineralocorticoid replacement therapy is indicated among patients with evidence of primary AI or after bilateral adrenalectomy. In these cases, fludrocortisone should be initiated when hydrocortisone dosages fall below 50 mg/day. Fludrocortisone is usually administered at dosages of 50–100 µg/day. Mineralocorticoid replacement is monitored based on the development of clinical symptoms like salt cravings, volume depletion, and orthostatic hypotension, followed by measurement of renin activity levels to a target goal in the upper end of the reference range without development of side effects [106].

Preceding discharge of patients with a confirmed diagnosis of AI, education should be provided for early recognition of adrenal crisis and sick day rules. Patients should be equipped with a steroid emergency card to be placed in wallets, set up on smartphones, and a medical alert bracelet [106].

### 4.2. Renal Failure 

Renal injury is a rare complication associated with PPGL resection. The mechanism of renal injury is due to massive catecholamine release in the postoperative period that can potentially lead to: (1) Stimulatory effects on renin activity and (2) hypertensive crisis from severe vasoconstriction leading to hypoperfusion at the renal bed, ischemia, and necrosis of skeletal muscles provoking rhabdomyolysis [110,111]. In contrast, hypotension due to a rapid drop in catecholamine levels, or intravascular volume depletion, can lead to acute tubular necrosis [111].

Occasionally, “mass effect” from tumor may lead to renal ischemia causing direct compression of the renal artery or vasospasm secondary to catecholamines excess [112,113,114]. Renal artery stenosis has been reported only during a hypertensive crisis. Diagnosis can be established by the use of Doppler ultrasonography, gadolinium-enhanced 3D magnetic resonance angiography, and contrast-enhanced arteriography. Stenosis tends to be transient and reversible after tumor resection. Failure to correct the mass effect on the renal artery may lead to postoperative renal artery thrombosis, resulting in permanent kidney damage. A second surgery to correct renal artery stenosis is risky and might result in secondary nephrectomy [115]. In cases of persistent renal artery stenosis, percutaneous balloon angioplasty is recommended [116]. When angioplasty fails, open surgical revascularization should be attempted [117].

In scenarios of hypertension leading to acute kidney injury, antihypertensive therapy must be initiated as mentioned earlier in this review. In cases of severe rhabdomyolysis-related acute kidney injury, hemodialysis is recommended. Intravenous fluids must be used judiciously. Colloids, such as albumin 4%, are recommended in patients at risk or with pre-existent renal failure and low albumin levels. Nephrotoxic agents should be discontinued, and supportive care should be provided under nephrology guidance.

### 4.3. Hypoglycemia

PPGL patients can have glucose homeostasis abnormalities mediated by elevated catecholamine secretion leading to increased liver glycogenolysis, inhibited insulin secretion from pancreatic β-cells, and increased insulin resistance in the skeletal muscle. These pathological changes can lead to preoperative hyperglycemia [118]. Sudden withdrawal of plasma catecholamines and pre-existence of preoperative hyperglycemia may result in postoperative hypoglycemia [119]. Chen et al. reported other risk factors associated with postoperative hypoglycemia following PPGL resection including tumor size, higher pre-operative 24-h urine metanephrine levels, and prolonged operative time. Similarly, preoperative β-adrenoceptor blockers exposure leads to increased liver glycogenolysis subsequently contributing to hypoglycemia development [9].

Classic symptoms of hypoglycemia (anxiety, sweating, chills, irritability, lightheadedness, nausea, etc.) may be masked in the postoperative period due to residual effects of anesthesia, opioid, or β-adrenoceptor blocker use. If untreated, hypoglycemia may result in neuronal cell death and brain damage [120]. Consequently, serum glucose levels should be monitored at regular intervals for at least 24 h postoperatively following a PPGL resection [121]. If a patient develops hypoglycemia, evaluation of related risk factors (for example, associated medications, critical illness, sepsis, renal or hepatic failure, or adrenal insufficiency, etc.) and identification of reversible culprits are recommended, independent of the catecholamine levels and the surgical approach [122].

Treatment considerations include administration of glucose tablets, glucose gels, or carbohydrate containing juices to provide 15–50 g of glucose. Moreover, if the patient is unable to tolerate oral intake, treatment with Dextrose 5% infusion should be started and titrated to a glucose goal of >100 mg/dL. In emergent situations, where treatment with oral or intravenous dextrose is not feasible, administration of 1 mg intramuscular glucagon should be considered [123,124,125]. Institutional hypoglycemia treatment guidelines and hospital policies should be promptly enforced, depending on individual patient needs and access to available resources [121,126,127,128,129].

### 4.4. Intestinal Pseudo-Obstruction

Increased catecholamine levels in the postoperative period following PPGL resection affects gastrointestinal smooth muscle cells and inhibits acetylcholine release from the parasympathetic nervous system, resulting in complications ranging from transient intestinal motility abnormalities to constipation, pseudo-obstruction, bowel infarction and perforation [130,131]. Moreover, commonly used medications in the postoperative period like analgesics and CCBs might demonstrate these symptoms. Moreover, α-adrenoceptor stimulation induces vasoconstriction of mesenteric arteries leading to ischemic colitis, ulceration, necrosis, and intestinal perforation, particularly in patients with risk factors for atherosclerotic or microvascular disease such as diabetes mellitus [132,133]. Ischemic colitis may be transient and reversible or associated with increased morbidity involving full thickness of the bowel wall, causing infarction and irreversible stricture requiring segmental resection. Paralytic ileus presents as constipation, abdominal distension, and discomfort for more than two to three days postoperatively. Therefore, patients are encouraged to be mobile, constantly change position, and recommended to be in sitting posture soon after surgery. Additionally, pain tolerance should be monitored to reduce analgesic dosage and avoid opioid drugs as soon as possible. Oral intake is slowly advanced starting from liquids to semi-solid and finally solid diet. High fiber diet is supplemented as the patient tolerates an oral diet. However, the use of liquid laxatives (milk of magnesium, magnesium citrate, Miralax) or rectal suppositories (bisacodyl, docusate sodium, polyethylene glycol) is advisable in some patients to relieve discomfort. Conservative management is preferred for patients with colonic distension measured on a plain radiograph as <12 cm, which includes fasting, nasogastric suction, intravenous fluid and electrolytes replacement, and discontinuation of drugs affecting colonic motility (narcotics) [134]. If conservative treatment is inefficient, endoscopic desufflation and pharmacologic treatment are initiated, especially for those who are confirmed to have increased catecholamine levels due to widespread metastatic disease [135]. α-adrenoceptor blockers such as phenoxybenzamine or doxazosin are initially considered as the pharmacological management among those with widespread metastatic disease, due to their additional beneficial effect on the smooth muscle cells of the intestine and blood vessels. Moreover, Metyrosine, a tyrosine analog competitively inhibiting tyrosine hydroxylase (enzyme catalyzing rate-limiting step of conversion of tyrosine to dihydroxyphenylalanine DOPA in catecholamine synthesis) causes significant catecholamine store depletion inside tumor cells. Therefore, it is our strong recommendation to start metyrosine at 250 mg orally twice daily and, if necessary, increase the dose every 48 h. Occasionally, pseudo-obstruction is extremely severe with no improvement, despite using conservative and pharmacologic management. At this point of time, the use of phentolamine (short-acting, competitive α_1_- and α_2_-adrenoceptor antagonist) at a dose of 1–5 mg intravenously is justified. Additionally, phentolamine is also helpful in controlling elevated blood pressure following PPGL resection [134,136]. Nevertheless, major drawbacks associated with its use are: (1) Recurrence of pseudo-obstruction following discontinuation of drug and (2) intravenous administration requiring continuous intensive care monitoring to avoid a precipitous drop in blood pressure [137,138,139,140] (Table 5).

## 5. Other Common Surgical Complications

Complications observed after any surgical procedure which were not mentioned earlier include nausea, vomiting, urinary retention, hemorrhage, and wound infection. These complications are not elaborately described here as they can be managed similarly to any other surgical procedures. Meticulous monitoring along with the skills of a multidisciplinary team of physicians and appropriate nursing care, together with patient cooperation are helpful for faster recovery with no or minimal complications in the postoperative period.

PPGL patients are at a significant risk of bleeding, which is difficult to be identified as hypotension. It is not uncommon after tumor resection. Elevated blood pressure as a result of higher catecholamines causes hemorrhage either intra or postoperatively. Precise surgical technique is crucial to avoid redundant blood loss intraoperatively. Therefore, an experienced surgeon is preferred to resect PPGL, minimize blood loss, and make use of meticulous surgical techniques to accurately scissor out tumor from a complex site. However, in the incidence of major hemorrhage, hemodynamic stability of the patient is assessed, and appropriate transfusion is given as per the needs of the patient and clinical judgment of the surgeon and anesthetist. Depending on the risk to benefit ratio, necessary medications would be stopped or continued during the perioperative period and if in doubt, a consult specialist opinion is considered.

Surgical site/wound infection is a potential cause of morbidity and mortality in the postoperative period. Risk factors depend on location, nature of surgical wound/incision, and the procedure performed [141]. Postoperatively, regular wound inspection, infection control, and strict hygiene (specifically hand hygiene and early removal of clips, sutures, drains, and foreign materials) minimize the risk of wound infection. However, patients with surgical site infection present with pain, swelling, redness, warmth, purulent wound discharge, or dehiscence. Such patients are managed with appropriate laboratory work and targeted empiric antibiotic therapy is initiated as soon as possible.

Urinary retention, commonly regarded as a minor and trivial complication by surgeons, might cause increased restlessness, confusion, and delirium [142]. A catecholamine surge in the postoperative period from PPGL resection inhibits detrusor contraction via α-adrenoceptor mediated increase in bladder outlet and proximal urethral tone. Moreover, residual anesthetic effects cause bladder atony by acting as smooth muscle relaxants and interfering with autonomic regulation of detrusor muscle tone. Furthermore, vasopressors used to treat postoperative hypotension promote urinary retention by their effects on β-adrenoceptor in bladder and α-adrenoceptor in the bladder neck and proximal urethra. Moreover, aggressive fluid administration to correct hypotension might cause overdistension of the urinary bladder resulting in urinary retention. Diagnosis is based on the patient complaining of discomfort, palpable bladder on examination, and ultrasound bladder scanning for rapid and accurate assessment of bladder volume [143,144]. Therefore, the first step in the management of urinary retention in the postoperative period is urethral catheterization. If a prolonged period of urinary retention is observed, the use of indwelling catheter is not advised as it may result in infection. Henceforth, pharmacotherapy with α-adrenoceptor blockers such as tamsulosin, alfuzosin, or long-acting doxazosin is recommended.

Splenectomy is required among those patients who are undergoing unilateral adrenalectomy on the left side due to the presence of a large-sized PHEO. Such patients need to be vaccinated preoperatively against pneumococcus, *Hemophilus influenzae*, and Meningococcus [2].

## 6. Conclusions

Patients with PPGL resection must be managed appropriately in the intensive care setting during the postoperative period. Detailed physical examination and complete laboratory workup must be conducted at regular intervals to identify a patient at risk and provide treatment at the right point of time.

## Figures and Tables

**Figure 1 cancers-11-00936-f001:**
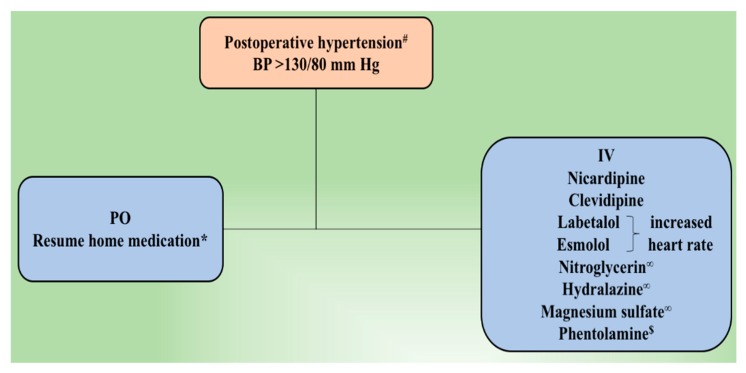
Postoperative management of hypertension following tumor resection. ^#^ Residual or metastatic disease causing an increased blood pressure is treated using α–adrenoceptor blocker. If necessary, β-adrenoceptor blocker and/or calcium channel blocker is added. β-adrenoceptor blocker might be used at first for epinephrine-secreting tumors. ^∞^ Management of hypertensive emergency. * Underlying essential hypertension is treated according to currently accepted guidelines. ^$^ Phentolamine is used to manage hypertensive crisis or in cases of resistant hypertension. BP, blood pressure; IV, intravenous; PO, per oral.

**Figure 2 cancers-11-00936-f002:**
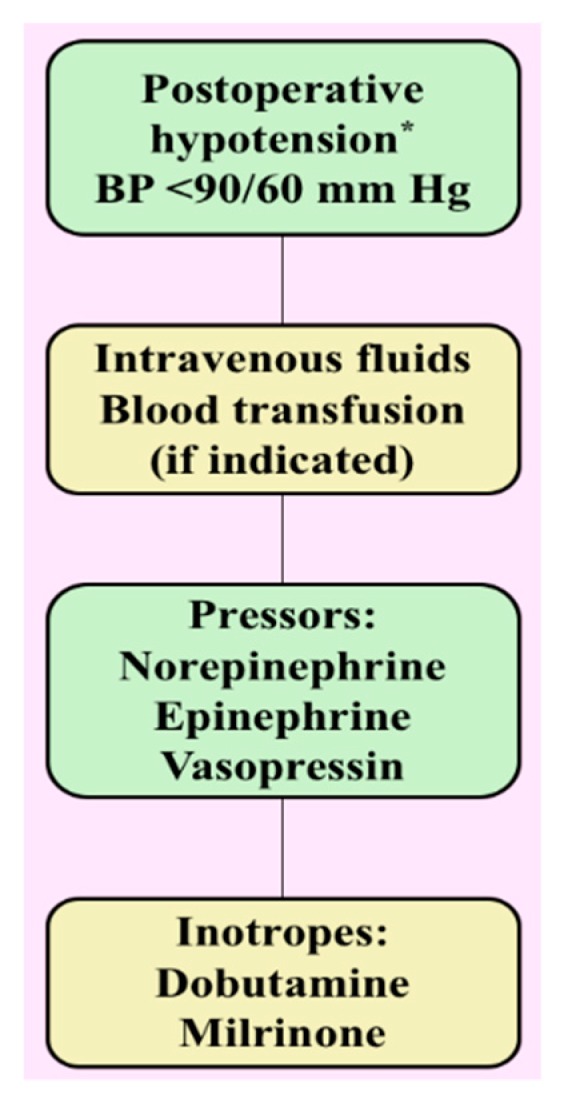
Postoperative management of hypotension following tumor resection. * In the differential diagnosis of hypotension consider downregulation of adrenoceptors, cardiogenic shock, sepsis, and medication-induced. BP, blood pressure.

**Figure 3 cancers-11-00936-f003:**
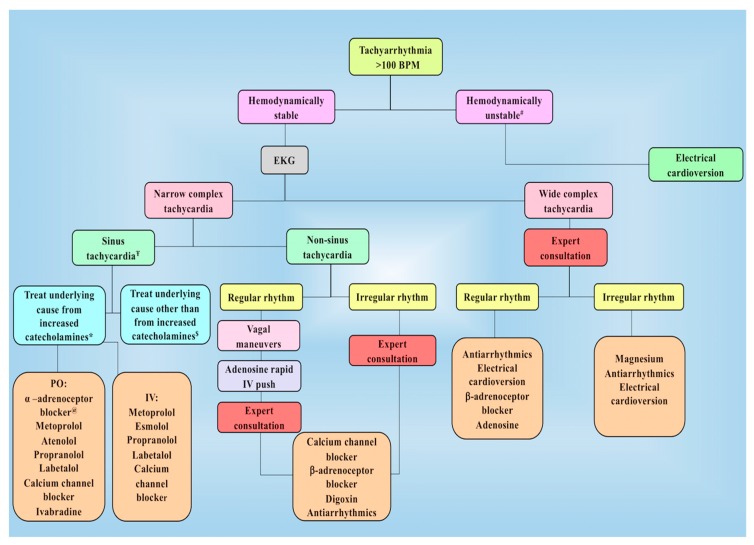
Management of tachyarrhythmia following PPGL resection in the postoperative period. ^#^ Unstable tachyarrhythmia implies patient has tachyarrhythmia along with hemodynamic instability or concerning symptoms. ^Ŧ^ Treatment in intensive care unit usually begins with calcium channel blockers. If heart rate is not well controlled, β-adrenoceptor blockers such as esmolol or metoprolol is added. In patients with increased blood pressure along with increased heart rate, use of combined α– and β-adrenoceptor blocker such as labetalol is recommended. Moreover, β-adrenoceptor blocker might be used at first for epinephrine-secreting tumors. * Underlying causes for increased catecholamines include their release during manipulation / resection of tumor and residual/ metastatic disease (patient must be on appropriate adrenoceptor blockade). ^$^ Other causes include inotropes used to correct postoperative hypotension, rebound tachycardia by discontinuation of β-adrenoceptor blockers used preoperatively as well as anemia, hypovolemia, pain, and anxiety. ^@^α-adrenoceptor blocker are used in the presence of residual or metastatic disease. BPM, beats per minute; EKG, electrocardiogram; IV, intravenous; PO, per oral.

**Figure 4 cancers-11-00936-f004:**
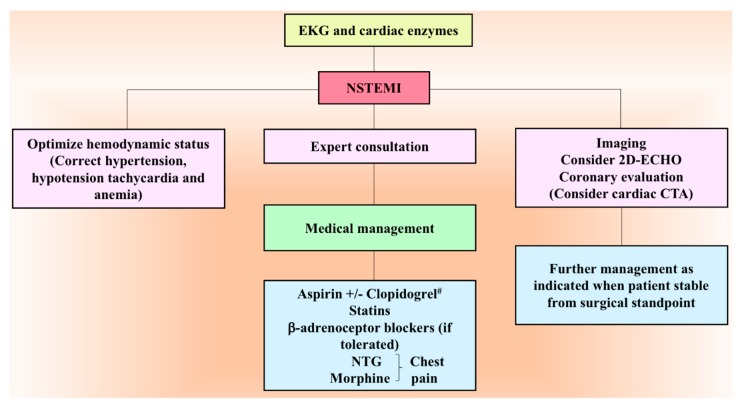
Management algorithm of postoperative NSTEMI. ^#^ Aspirin and clopidogrel is given in the postoperative period only when it is safe from surgical point of view. CTA, computed tomography angiography; 2D-ECHO, two-dimensional echocardiography; EKG, electrocardiography; NSTEMI, non-ST-elevation myocardial infarction; NTG, nitroglycerin.

**Table 1 cancers-11-00936-t001:** Characteristics of subtypes of adrenergic receptors.

Adrenoceptor Subtype	Agonists	Tissue	Responses
α_1_ *	EPI≥NE>>Iso Phenylephrine	Vascular smooth muscle	Vasoconstriction
		Liver ^Ŧ^	Glycogenolysis, gluconeogenesis
		Intestinal smooth muscle	Hyperpolarization and relaxation
		Heart	Chronotropic, arrhythmias
α_2_ *	EPI≥NE>>Iso Clonidine	Pancreatic islets (/cells)	Decreased insulin secretion
		Platelets	Aggregation
		Nerve terminals	Decreased release of NE
		Vascular smooth muscle	Vasoconstriction
β_1_	Iso>EPI=NE Dobutamine	Heart	Chronotropic and inotropic
β_2_	Iso>EPI>>NE Terbutaline	Juxtaglomerular cells Smooth muscle (vascular, bronchial, and gastrointestinal) Skeletal muscle Liver ^Ŧ^	Increased renin secretion Relaxation
β_3_ ^ŧ^	Iso=NE>EPI	Adipose tissue	Glycogenolysis, uptake of K^+^ Glycogenolysis, gluconeogenesis lipolysis

* At least three subtypes each of α_1_– and α_2_–adrenoceptor are known, but distinctions in their mechanisms of action and tissue locations have not been clearly defined. ^Ŧ^ In some species (e.g., rat), metabolic responses in the liver are mediated by α_1_–adrenoceptor, whereas in others (e.g., dog), β_2_–adrenoceptor are predominantly involved. Both types of receptors appear to contribute to responses in humans. ^ŧ^ Metabolic responses in adipocytes and certain other tissues with atypical pharmacologic characteristics may be mediated by this subtype of receptor. Most β_2_–adrenoceptor antagonists (including propranolol) do not block these responses. EPI, epinephrine; NE, norepinephrine; Iso, isoproterenol; AV, atrioventricular. Adapted from Goodman and Gillman’s The Pharmacological Basis of Therapeutics [13].

**Table 2 cancers-11-00936-t002:** Responses of effector organs to autonomic nerve impulses.

Effector Organs	Adrenergic Impulses		Cholinergic Impulses
	Receptor Type *	Responses ^Ŧ^	Responses ^Ŧ^
*Heart ^ŧ^*			
SA node	β_1_, β_2_	Chronotropic ++	Chronotrophy −−, vagal arrest +++
Atria	β_1_, β_2_	Inotropic and chronotropic ++	Inotropic −−, shortened AP duration ++
AV node	β_1_, β_2_	Increase in automaticity and chronotropic ++	Chronotropic −−, AV block +++
His-Purkinje system	β_1_, β_2_	Increase in automaticity and chronotropic +++	Little effect
Ventricles	β_1_, β_2_	inotropic, chronotropic automaticity, and rate of idioventricular pacemakers +++	Slight decrease in contractility
*Arterioles*			
Coronary	α_1_, α_2_, β_2_	Constriction +, dilations ^§^ ++	Constriction +
Skin and mucosa	α_1_, α_2_	Constriction +++	Dilation ^||^
Skeletal muscle	α_1_, β_2_	Constriction +, dilation ^§^ ++, ^$^++	Dilation **+
Cerebral	α_1_	Constriction (slight)	Dilation ^||^
Pulmonary	α_1_, β_2_	Constriction +, dilations ^§^	Dilation ^||^
Abdominal viscera	α_1_; β_2_	Constriction +++, dilation ^$^+	−
Salivary glands	α_1_, α_2_	Constriction +++,	Dilation++
Renal	α_1_, α_2_, β_1_, β_2_	Constriction +++, dilation ^$^+	−
*Veins (systemic)*			
Veins	α_1_, α_2_, β_2_	Constriction ++, dilation++	−
*Lung*			
Tracheal and bronchial muscle	β_2_	Relaxation +	Contraction ++
Bronchial glands	α_1_, β_2_	Decreased secretion, increased secretion	Stimulation +++
*Stomach*			
Mobility and tone	α_1_, α_2_, β_2_	Decrease (usually) ^ŦŦ^+	Increase +++
Sphincters	α_1_	Contraction (usually) +	Relaxation (usually) +
Secretion		Inhibition	Stimulation +++
*Intestine*			
Mobility and tone	α_1_, α_2_, β_1_, β_2_	Decrease (usually)+	Increase +++
Sphincters	α_1_	Contraction (usually) +	Relaxation (usually) +
Secretion	α_2_	Inhibition	Stimulation ++
*Gallbladder and ducts*			
	β_2_	Relaxation +	Contraction +
*Kidney*			
Renin Secretion	α_1_, β_1_	Decrease +, increase ++	−
*Urinary bladder*			
Detrusor	β_2_	Relaxation (usually) +	Contraction +++
Trigone and sphincter	α_1_	Contraction ++	Relaxation ++
*Ureter*			
Mobility and tone	α_1_	Increase	Increase (+)
*Adrenal medulla*			
		−	Secretion of epinephrine and norepinephrine (primarily nicotinic and secondarily muscarinic)
*Skeletal muscle*			
	β_2_	Increased contractility, glycogenolysis, K^+^ uptake	−
*Liver*			
	α_1_, β_2_	Glycogenolysis and gluconeogenesis ^$$^ +++	−
*Pancreas*			
Acini	α	Decreased secretion +	Secretion ++
Islets (β cells)	α_2_	Decreased secretion +++	−
	β_2_	Increased secretion +	−

* Where a designation of subtype is not provided, the nature of the subtype has not been determined unequivocally. ^Ŧ^ Responses are designated highest (+++ and −−) to lowest (+ and −) to provide an approximate indication of the importance of adrenergic and cholinergic nerve activity in the control of the various organs and functions listed. ^ŧ^ Although it had been thought that β_1_–adrenoceptor predominates in the human heart. Recent evidence indicates some involvement of β_2_–adrenoceptor. ^§^ Dilation predominates in situ due to metabolic autoregulatory phenomena. ^||^ Cholinergic vasodilation as these sites is of questionable physiologic significance. ^$^ Over the usual concentration range of physiologically released, circulating epinephrine, β–adrenoceptor response (vasodilation) predominates in blood vessels of skeletal muscle and liver, α–receptor response (vasoconstriction), in blood vessels of other abdominal viscera. The renal and mesenteric vessels also contain specific dopaminergic receptors, activation of which causes dilation. ** Sympathetic cholinergic system causes vasodilation in skeletal muscle, but this is not involved in most physiologic responses. ^ŦŦ^ It has been proposed that adrenergic fibers terminate at inhibitory β–adrenoceptors on smooth muscle fibers and at inhibitory α–adrenoceptors on parasympathetic cholinergic (excitatory) ganglion cells of Auerbach’s plexus. ^$$^ There is significant variation among species in the type of receptors that mediates certain metabolic responses. α- and β-adrenoceptor responses have not been determined in human beings. A β_3_–adrenoceptor has been cloned and may mediate lipolysis or thermogenesis or both in fat cells in some species; SA, sinoatrial; AV, atrioventricular. Adapted from Goodman and Gillman’s The Pharmacological Basis of Therapeutics [13].

**Table 3 cancers-11-00936-t003:** Anti-hypertensive medications used in hypertensive crisis after pheochromocytoma and paraganglioma (PPGL) resection.

Anti-Hypertensive Medication	Mechanism of Action	Route of Administration	Dose
α- and β-adrenoceptor blockers			
Phentolamine	Competitive α_1_- and α_2_-adrenoceptor blocker	IV	Bolus dose 5 g Maintenance dose 0.5 mg/min
Esmolol	β_1_-adrenoceptor antagonist	IV	Starting dose 0.5 ml/kg Maintenance dose 50–300 µg/kg/min
Metoprolol	β_1_-adrenoceptor antagonist	IV	5 mg every 5 mins as tolerated up to 15 mg total dose
Labetalol	Selective α_1_- and nonselective β-adrenoceptor antagonist	IV	Loading dose 10–20 mg, double initial dose every 10 mins until target blood pressure is attained
Calcium channel blockers			
Nicardipine	NE mediated transmembrane calcium influx into vascular smooth muscles	IV	Starting dose 5 mg/h, dose increased by 2.5 mg/h every 5mins to a maximum of 15 mg/h
Clevidepine	Increase cardiac output Decrease afterload	IV	Starting dose 1–2 mg/h Maintenance dose 4–6 mg/h
Others			
Nitroglycerin	Venous dilator Decrease preload	IV	Starting dose 5–10 µg/min (increase the dose by 5 µg/min every 5 mins until desired effect is achieved)
Hydralazine	Decrease arterial vascular resistance	IV	10 mg over 2 mins, additional doses as needed
Magnesium sulfate	•Inhibition of release catecholamines •Direct inhibition of catecholamines receptors •Endogenous calcium antagonist	IV	Bolus dose 2–4 g over 2 mins Maintenance dose 1–2 g/h, dose adjusted on magnesium blood levels

Abbreviations: IV, intravenous; NE, norepinephrine.

**Table 4 cancers-11-00936-t004:** Medical Therapy for NSTEMI following PPGL resection.

Medications	Dosage	Goal of Treatment
Aspirin	325 mg PO	Inhibition of platelet aggregation and activation
High-intensity Statin		Lowers LDL cholesterol levels in blood by approximately ≥ 50%, atherosclerotic plaque stabilization
Atorvastatin	40–80 mg PO	
Rosuvastatin	20–40 mg PO	
β-adrenoceptor blocker		Lowering heart rate, pain resolution, ST-segment normalization
Metoprolol	1–5 mg IV, incrementally repeat as needed up to 15 mg total dose 25–50 mg PO three to four times a day	
Esmolol	10–50 mg IV bolus, infusion up to 200 µg/kg/min	
Morphine sulfate	2–5 mg IV, repeat as needed	Pain control
Nitroglycerin	0.4 mg sublingual every five minutes up to a total of three doses. Transdermal patch starting at 0.2 mg/h and increasing to 0.6 mg/h with drug-free period from 6 to 8 PM 50 µg/min IV, titrate upwards as mean arterial pressure tolerates	Pain elimination by coronary vasodilation and ST-segment normalization

Abbreviations: BPM, beats per minute; IV, intravenous; LDL, low density lipoprotein; PO, per oral.

**Table 5 cancers-11-00936-t005:** Postoperative complications following PPGL resection.

Complication	Reason	Recommended First Line Management
Hypertension	•Incomplete tumor removal •Tumor present at unknown location •Metastatic tumor •Excessive vasopressor use •Surplus IV fluids administration •Pain medication •Underlying essential hypertension	Nicardipine Labetalol
Hypotension	•Chronically low circulating plasma volume •Prolonged (preoperative) α-adrenoceptor blockade action •Abrupt decrease in serum catecholamines levels •Downregulation of adrenoceptors •Blood loss	IV fluids ^#^
Arrhythmia	Tachyarrhythmia: •Elevated sympathetic activity from increased catecholamines levels and pain •Use of inotropes for postoperative hypotension •Rebound effect of preoperative discontinuation of β-adrenoceptor blockers	*Stable narrow complex sinus tachycardia:* Treat underlying cause *Stable regular narrow complex non-sinus tachycardia*: Vagal maneuvers- adenosine, expert consultation *Stable irregular narrow complex tachycardia*: Expert consultation. *Stable wide complex tachycardia*: Expert consultation *Hemodynamically unstable tachycardia*: Electrical cardioversion
	Bradyarrhythmia: •Progression of native conduction disease •Electrolyte disturbance •Sinus node dysfunction •Heart block •Excessive medication used such as β-adrenoceptor blockers	Treatment of underlying abnormalities Atropine Dopamine Epinephrine
Myocardial infarction	Increased catecholamines causes myocyte injury by: •Hemodynamic compromise •Tachycardia •Increased oxygen consumption •Coronary artery vasoconstriction	12-lead EKG Expert consultation
Heart failure	•Desensitized adrenoceptors on myocardium •Cardiomyopathy	Expert consultation
Cerebrovascular accident	•Uncontrolled hypertension •Thrombotic or embolic occlusion of cerebral artery •Rupture of intracranial artery leading to hemorrhage	Expert consultation
Adrenocortical insufficiency	•PHEO resection with concomitant cortisol hypersecretion	Hydrocortisone Fludrocortisone
	•Bilateral adrenalectomy	
Renal Failure	•Hypoperfusion of renal bed (hypotension, hypertension and massive bleeding) •Secondary to rhabdomyolysis	Antihypertensive medication (if hypertension exits) IV fluid therapy based on electrolytes Hemodialysis
Hypoglycemia	•Hyperinsulinemia from increased catecholamine secretion (predominantly β-adrenergic) •Sudden withdrawal of catecholamines	50% Dextrose (0.5 ml ampules) Maintenance fluid must include 5–20% dextrose.
Intestinal pseudo-obstruction	•Hypomotility from increased catecholamines •Mesenteric vasoconstriction (predominantly α-adrenergic) •Use of opioid analgesics	Laxatives diet with high fiber content, enema

^#^ Blood transfusion if indicated. Abbreviations: BSA, body surface area; EKG, electrocardiogram; IV, intravenous; PHEO, pheochromocytoma.

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
