# Peer review of "Postoperative Management in Patients with Pheochromocytoma and Paraganglioma"

_cancers, 2019, doi:10.3390/cancers11070936_

Reviewer 1 Report

This comprehensive review by Mamilla and co-workers is an elaborate review of current clinical aspects regarding postoperative management of PPGL patients. The manuscript is structured, well-written with a nice flow, and the authors have balanced the study well between clinical management/practical issues and the physiological background/biological mechanisms. The work is a welcomed addition to the field in general. Some points to consider:

Page 1, row 15. "orresponding author". Please correct.

It is not entirely clear to me what Figure 1 is thought to simplify/represent, and I lack a structured Figure Legend with a brief summary of what this scheme is thought to demonstrate. The same goes for Figure 2.

Page 15, rows 500-501. I think the authors could consider to discuss the modified NIHSS (PMID: 19689755) in this aspect, as compared to the conventional NIHSS.

Page 20, rows 679-680. Is splenectomy recommended for any large sized left-sided pheo, or only when the tumor cannot be easily resected with negative margins without a splenectomy?

Author Response

Dear Reviewer,

Reviewer 1. Comments for the Author.

This comprehensive review by Mamilla and co-workers is an elaborate review of current clinical aspects regarding postoperative management of PPGL patients. The manuscript is structured, well written with a nice flow and the authors have balanced the study well between clinical management/practical issues and the physiological background/biological mechanisms. The work is a welcomed addition to the field in general. Some points to consider:

      We would like to thank the Reviewer for his/her review and comments. We have tried to reply to all the questions accordingly and very much appreciate the effort of the Reviewer in helping us improve our manuscript.

1.      Page 1, row15. “orresponding author. Please correct.

We thank the Reviewer for bringing this to our attention. We apologize for this error. Corresponding author has been corrected as suggested. 

2.      It is not entirely clear to me what Figure 1 is thought to simplify/represent, and I lack a structured legend with a brief summary of what this scheme is thought to demonstrate. The same goes for figure 2.

We thank the Reviewer for bringing this to our attention. We regret for not being absolutely clear for Figure 1 and Figure 2, causing confusion to the readers and Reviewers. 

Figure 1 describes that some patients in the postoperative period develops hypertension especially while transferring to the intensive care unit (hypertension could be caused by pain, pain medication, residual tumor, etc.). Therefore, some antihypertensive medications could be considered depending on the degree of hypertension. We wanted to show that there are two available options to manage hypertension in the postoperative period following PPGL resection with either peroral i.e. to resume home medications or medications to be administered in the intensive care unit in the box showing intravenous medications (based on our institutional experience suggesting which medication is preferred, for e.g. Nicardipine has been found to be very useful in such setting).

The same applies to Figure 2. indicating postoperative management of hypotension following tumor resection where we usually give intravenous fluids at first as the patient may be:

a)      Volume depleted

b)      Still on-board adrenoceptor blockers

c)      Sudden decrease in catecholamine concentration (due to tumor removal), therefore, causing profound vasodilation.   

3.      Page 15, rows 500-501. I think the authors could consider to discuss the modified NIHSS (PMID: 19689755) in this aspect, as compared to the conventional NIHSS.

We thank the Reviewer for this very clinically useful comment and bringing this specific study to our attention.

We have added the modified NIHSS used to perform detailed neurological examination in patients with increased risk of stroke [1].

4.      Page 20, rows 679-680. Is splenectomy recommended for any large sized left-sided pheo, or only when the tumor cannot be easily resected with negative margins without a splenectomy?

We thank the reviewer for asking this important question. Yes, splenectomy is recommended only when the tumor cannot be resected.

Reviewer 2 Report

Dear Editor, 

This is a very interesting and original review about the postoperative management in patients with Pheochromocytoma and Paraganglioma and their potential complications. Thus this paper could be a useful guide for all clinicians and surgeons.

Some minor comments/suggestions which could be included in the manuscript (if the data are available) are the following: 

-Line 450: Authors state that: “Clinical presentation of myocardial infarction in the postoperative period is often asymptomatic or non-specific…”. Please could they clarify why is often asymptomatic? (i.e use of analgesics as mentioned in the next phrase… other reasons?)

-Line 572: Authors state that: “In cases of rhabdomyolysis related acute kidney injury, hemodialysis is recommended.”  Is this true in all cases of rhabdomyolysis due to pheochromocytoma/paraganglioma?

-Despite the rarity of pheochromocytoma/paraganglioma in pregnancy maybe a separate  paragraph should be dedicated for these cases. 

-It could be useful if authors state in few lines which drugs should be avoided post-operatively for the management of each complication (ie because they may cause false-positive results for catecholamines measurement (methyl-dopa) or because they are contraindicated particularly in these patients). 

-Are there any epidemiological data about the incidence of myocardial infarction, cerebrovascular accident or heart failure postoperatively in these patients?

-Patients with large tumors but no detectable catecholamines excess pre-operatively should they also be monitored with precaution post-surgery for hypertension, hypotension, heart failure etc and if yes why ?

Author Response

Dear Reviewer,

Reviewer 2. Comments for the Author.

This is a very interesting and original review about the postoperative management in patients with Pheochromocytoma and Paraganglioma and their potential complications. Thus this paper could be a useful guide for all clinicians and surgeons. Some minor comments/suggestions which could be included in the manuscript (if the data are available) are the following: 

      We would like to thank the Reviewer for his/her review and comments. We have tried to respond to all the questions accordingly and very much appreciate the effort of the Reviewer in helping us improve our manuscript.

1.      Line 450: Authors state that: “Clinical presentation of myocardial infarction in the postoperative period is often asymptomatic or non-specific…”. Please could they clarify why is often asymptomatic? (i.e use of analgesics as mentioned in the next phrase… other reasons?)

We thank the Reviewer for careful and in-depth review of the article. The reason behind asymptomatic presentation of myocardial infarction in the postoperative period are:

a)      Classical clinical approaches i.e. chest pain, EKG changes, and increased cardiac enzymes are misleading, thus, missing acute coronary events requiring immediate therapeutic intervention

b)      Non-specific EKG modifications following surgery can obscure the clinical presentation

c)      Elevation in cardiac enzymes following surgery complicates the interpretation of these biomarkers in the diagnosis of myocardial infarction

2.      Line 572: Authors state that: “In cases of rhabdomyolysis related acute kidney injury, hemodialysis is recommended.”  Is this true in all cases of rhabdomyolysis due to pheochromocytoma/paraganglioma?

We thank the Reviewer for asking this important question regarding our article. The mainstay of treatment of rhabdomyolysis is restoration of adequate renal perfusion and increasing urine flow, thus, preventing and treating acute kidney injury. In severe cases, i.e. when a patient develops acute kidney injury with laboratory investigations reporting hyperkalemia, metabolic acidosis and volume overload, renal replacement therapy is indicated. Therefore, the use of hemodialysis depends on the severity of injury in PPGL patients.

Upon Reviewers suggestion, we have modified this section. 

3.      Despite the rarity of pheochromocytoma/paraganglioma in pregnancy maybe a separate paragraph should be dedicated for these cases.

We thank the Reviewer for bringing this to our attention. We agree that pregnancy and PPGL is a very important topic to discuss. However, due to some limitations in the length of the manuscript, as well as our limited experience with pregnant woman and PPGL treatment (due to NIH and our protocol policy of not including pregnant woman and performing there surgical treatment at NIH), we have opted not to include this part in the manuscript. Nevertheless, we feel this is extremely important topic for future review to be done in collaboration with colleagues who have experience with pregnant woman with PPGL. Thank you so much for the suggestion. 

4.      It could be useful if authors state in few lines which drugs should be avoided post-operatively for the management of each complication (i.e. because they may cause false-positive results for catecholamines measurement (methyl-dopa) or because they are contraindicated particularly in these patients). 

We thank the Reviewer for bringing this very interesting suggestion. We agree with the Reviewer that there are some drugs that should be avoided postoperatively in patients following tumor resection. However, it depends on the type of complication detected in these patients. Nonetheless, the drugs that would cause false positive catecholamines and metanephrines measurements are not important to be considered because catecholamines and metanephrines should not be measured postoperatively (during the first 2-3 weeks following surgical tumor resection). This is due to the fact that many postoperative conditions (pain, pain medication, infection, hemorrhage etc.) as well as drugs may affect catecholamine and metanephrine levels. Moreover, the review of drugs which provokes release of catecholamines and interfere with their turnover are contraindicated in patients with PPGL (same applies to those patients where residual tumor is present) is already published (PMID: 17989126). Therefore, we have not discussed this in our article.

5.      Are there any epidemiological data about the incidence of myocardial infarction, cerebrovascular accident or heart failure postoperatively in these patients?

We thank the Reviewer for asking an excellent and challenging question. At present time there is no available epidemiological data related to these complications seen in the postoperative period following PPGL resection. However, we estimate that this could be about 5%-10%, especially at centers that have not good exposure with these tumors.

6.      Patients with large tumors but no detectable catecholamines excess pre-operatively should they also be monitored with precaution post-surgery for hypertension, hypotension, heart failure etc. and if yes why?

We thank the Reviewer for asking extremely clinically interesting question. Patients with large tumors but no detectable catecholamines excess preoperatively need not to be monitored because they are biochemically silent. However, surgery itself can result in hemorrhage, sepsis, organ failure etc. Therefore, monitoring should be done in same way as for patients who undergo any other surgery along with intensive care unit care in the postoperative period.

1.         Meyer, B.C. and P.D. Lyden, The modified National Institutes of Health Stroke Scale: its time has come. Int J Stroke, 2009. 4(4): p. 267-73.